

# Comparing the Postural Assessment Scale for Stroke and Berg Balance Scale for predicting community walking ability at discharge in subacute stroke: a prospective cohort study

Chutipa Worraridthanon[1,2], Maria Justine[1,3] and Akkradate Siriphorn[1]

[1] Department of Physical Therapy, Faculty of Allied Health Sciences, Chulalongkorn University, Bangkok, Bangkok, Thailand
[2] Department of Physical Therapy, Sirindhorn National Medical Rehabilitation Institute, Nonthaburi, Thailand
[3] Centre for Physiotherapy Studies, Faculty of Health Sciences, Universiti Teknologi MARA, Puncak Alam, Selangor, Malaysia

Corresponding author
Akkradate Siriphorn,
akkradate.s@chula.ac.th

## ABSTRACT

**Background:** Balance assessment is crucial for predicting community ambulation outcomes in subacute stroke patients undergoing rehabilitation. This study aims to compare the accuracy of the Postural Assessment Scale for Stroke Patients (PASS) and the Berg Balance Scale (BBS) in predicting community walking ability at discharge from rehabilitation.

**Methods:** This prospective cohort study included 47 stroke patients admitted to a 4-week inpatient rehabilitation program. Patients were assessed with PASS and BBS at admission. Discharge assessments included the Functional Ambulation Categories and 6-Min Walk Distance tests. Statistical analysis involved calculating the area under the receiver operating characteristic curve (AUC), sensitivity, specificity, likelihood ratios, and cut-off scores.

**Results:** PASS and BBS demonstrated excellent predictive accuracy, with AUC values of 0.955 (95% CI [0.850–0.994]) for PASS and 0.991 (95% CI [0.906–1.000]) for BBS. Cut-off scores were >28 for PASS and >46 for BBS. Sensitivity was high for both (94.44%, 95% CI [72.7–99.9]), while BBS had superior specificity (96.43%, 95% CI [81.7–99.9]) compared to PASS (85.71%, 95% CI [67.3–96.0]). BBS also had a higher positive likelihood ratio (26.44 *vs.* 6.61). The difference in AUC values was non-significant ($p$ = 0.093).

**Conclusions:** PASS and BBS assessed at admission are highly accurate tools for predicting community ambulation at discharge in subacute stroke patients, with BBS demonstrating a slight advantage, particularly in its positive predictive value. These findings support the use of both scales to guide rehabilitative clinical decision-making.

## INTRODUCTION

Stroke is a leading cause of long-term disability worldwide, often resulting in significant mobility impairments and dependence on caregivers for daily activities (*Katan & Luft, 2018*; *Lee et al., 2024*). These challenges are primarily due to a combination of motor weakness, sensory deficits, motor incoordination, spasticity, and balance issues (*Lee, 2019*). Previous studies have shown that balance and trunk control are key predictors of post-stroke walking ability, making the selection of appropriate balance assessment tools crucial for optimizing rehabilitation (*Preston et al., 2021*).

Three widely used tools for balance assessment in stroke rehabilitation are the Berg Balance Scale (BBS), the Postural Assessment Scale for Stroke Patients (PASS), and the Mini-Balance Evaluation Systems Test (Mini-BESTest). The BBS is commonly used for balance assessment, given its capacity to evaluate fall risk and functional stability through a series of tasks (*Blum & Korner-Bitensky, 2008*; *Inoue et al., 2023*). It measures both static and dynamic balance, providing comprehensive insights into a patient's functional stability, and typically takes about 15–20 min to administer (*Dos Santos et al., 2023*; *Inoue et al., 2023*). However, its utility is limited by floor and ceiling effects, which reduce its sensitivity for patients with severe impairments or near-complete recovery (*Blum & Korner-Bitensky, 2008*; *Louie & Eng, 2018*). In contrast, the PASS is specifically designed for patients with stroke and emphasizes postural control during static and dynamic tasks relevant to daily activities (*Benaim et al., 1999*; *Chien et al., 2007*; *Yoshimoto et al., 2016*). It is quicker to administer (10–15 min) and particularly suited for individuals with significant functional limitations, demonstrating predictive validity with scores above 12.5 correlating with independent ambulation (*Huang et al., 2016*). The Mini-BESTest is another commonly used balance assessment tool that focuses on dynamic balance components, including anticipatory postural adjustments, reactive postural control, sensory orientation, and dynamic gait (*Tsang et al., 2013*). It has demonstrated excellent reliability and predictive validity for fall risk across various neurological conditions, including stroke (*Tsang et al., 2013*). However, Mini-BESTest is more comprehensive and time-intensive, making it less practical for routine assessments in rehabilitation settings. Additionally, its primary focus on dynamic balance may not fully capture the postural control deficits that are critical for early-stage stroke rehabilitation. Given these considerations, this study compares the predictive accuracy of PASS and BBS for community walking ability at discharge.

While both tools offer valuable insights into balance and postural control, accurately predicting community ambulation after stroke rehabilitation remains challenging. Walking speed alone is insufficient to reliably predict community walking capability (*Alvarenga et al., 2023*; *Fulk et al., 2017*). Combining the 6-Minute Walking Distance (6MWD) and Functional Ambulation Categories (FAC) provides a more comprehensive evaluation. The 6MWD, a simple and safe measure of walking endurance, distinguishes between house-bound and community ambulators with a threshold of 205 meters (71% specificity, 79% sensitivity, 74% overall accuracy) (*Fulk & He, 2018*; *Fulk et al., 2017*; *Kubo et al., 2018*). The FAC, which categorizes ambulation across six levels based on assistance

required, complements this by assessing functional independence in walking (*Mehrholz et al., 2007*). However, neither tool fully accounts for balance deficits or other factors influencing community walking, highlighting the need for a multifaceted approach to prediction.

The subacute phase of stroke recovery, spanning from 7 days to 6 months post-onset, is a critical period for intervention, where timely and accurate assessments can guide targeted rehabilitation strategies to maximize functional outcomes (*Bernhardt et al., 2017*). While the BBS and PASS are widely used, there is limited evidence directly comparing their utility during this phase to predict walking ability at discharge. The PASS was specifically developed to assess both static and dynamic balance in patients with stroke. It is particularly effective in detecting balance improvements in individuals with severe balance deficits, an area where the BBS and Mini-BESTest may have limitations (*Huang et al., 2020*). The Mini-BESTest, though highly reliable and useful in assessing dynamic balance and fall risk, is more commonly used across various neurological conditions rather than specifically for post-stroke rehabilitation. Furthermore, PASS has demonstrated strong predictive validity for ambulation and functional independence, making it an appropriate choice for evaluating community walking ability post-stroke (*Huang et al., 2016*). Given these considerations, this study compares PASS and BBS to determine their predictive accuracy for community walking ability at discharge. The BBS's susceptibility to floor and ceiling effects contrasts with the PASS's strength in assessing patients with greater impairments, suggesting that their predictive accuracy may differ depending on the patient's functional level and stage of recovery.

Given the strengths and limitations of both tools, a comparative study is warranted to determine which tool more accurately predicts community walking ability in patients with subacute stroke at discharge from rehabilitation. By integrating balance assessments with established measures such as 6MWD and FAC, this study aims to provide evidence-based insights to enhance clinical decision-making, optimize tool selection, and ultimately improve rehabilitation outcomes for patients with stroke.

## MATERIALS AND METHODS

### Study design

This prospective cohort study assessed patients with stroke upon admission to a rehabilitation unit using the PASS and the BBS. Following these assessments, participants underwent a 4-week rehabilitation program consisting of 16 physical therapy sessions. Walking ability was classified using a combined approach incorporating the 6MWD test and FAC. In this study, community ambulators were defined as individuals who could walk independently outdoors and navigate public spaces, while home ambulators were those whose walking ability was primarily limited to their home environment due to mobility impairments or safety concerns. Participants who achieved ≥205 m on the 6MWD and FAC level 5 were classified as community ambulators, while those walking <205 m with FAC levels 0–4 were classified as home ambulators. Data collection was conducted in a tertiary care rehabilitation hospital specializing in stroke rehabilitation

between May and July 2024. Assessments at admission and discharge were conducted in dedicated therapy spaces, ensuring consistency in measurement conditions.

## Sample size calculation

The sample size was calculated using MedCalc software (version 19.1.5; MedCalc Software Ltd, Ostend, Belgium). An expected AUC of 0.8 was chosen because it falls within the moderate accuracy range (AUC 0.70–0.90), which is widely considered acceptable for predictive models in clinical research. Using this expected AUC, a significance level (alpha) of 0.05, and a power of 80% (beta = 0.20), the minimum sample size was estimated to be 36 participants. Accounting for a potential dropout rate of 30%, the final target sample size was set at 47 participants.

## Participants

This study included 47 patients with stroke from the Sirindhorn National Medical Rehabilitation Institute in Thailand, with data collected between May and July 2024. A total of 60 eligible patients were initially approached, of whom 47 agreed to participate and met the inclusion criteria. Eligibility criteria included: (1) men and women aged 18–80 years; (2) diagnosis of stroke confirmed by a neurologist using MRI or CT; (3) first stroke occurring between 2 and 6 months prior; (4) stable medical condition defined by resting vital signs within normal ranges (blood pressure: 90/60–140/90 mmHg; heart rate: 60–100 beats per minute; respiratory rate: 12–20 breaths per minute) and the absence of acute medical complications, such as infections, uncontrolled hypertension, or cardiac arrhythmias, as documented in the medical record and confirmed by the attending physician; and (5) ability to understand and follow instructions, demonstrated by successful completion of a two-step instruction task (*e.g.*, "Raise your arm and touch your head"). Exclusion criteria were: (1) presence of conditions such as severe osteoarthritis, recurrent stroke, vestibular disorders (including vertigo), or Parkinson's disease that could affect gait or balance; (2) history of knee or hip arthroplasty that could limit mobility; (3) leg pain or fatigue on the day of evaluation; and (4) completion of fewer than 12 rehabilitation sessions, representing less than 80% of the total program.

## Research protocol

This study protocol was approved by The Sirindhorn National Medical Rehabilitation Institute's Subcommittee on Human Research Ethics (No. 67014) and preregistered at www.thaiclinicaltrials.org (No. TCTR20240528005). All participants provided written informed consent prior to their participation. Initially, participants were assessed using PASS and BBS at admission to evaluate postural control and balance. To mitigate order effects, a random lottery determined the sequence of assessments, with a 10-min rest period between tests. Following these evaluations, participants underwent a structured 4-week rehabilitation program consisting of 16 sessions, each 60 min long, focusing on strength, balance, bed mobility, and ambulation.

At discharge, walking ability was classified using a combined approach incorporating the 6MWD and FAC. Participants achieving ≥205 m on the 6MWD and FAC level 5 were

classified as community ambulators, while those walking <205 m with FAC levels 0–4 were classified as home ambulators (*Fulk et al., 2017*; *Mehrholz et al., 2007*). Importantly, ambulatory classification was conducted only at discharge, as this study focused on evaluating the predictive accuracy of PASS and BBS rather than pre-rehabilitation walking status.

To minimize measurement bias, assessments at admission (PASS and BBS) and at discharge (6MWD and FAC) were conducted by different assessors. The admission assessor conducted pre-rehabilitation evaluations and was not involved in the discharge assessments, ensuring independence between the two phases. Similarly, the assessor at discharge was blinded to the admission assessments and scores, maintaining unbiased evaluations of the participants' walking ability at the end of rehabilitation. All assessors were licensed physical therapists with over 5 years of experience working with patients with stroke and were well-trained in administering the tools used in this study.

## Measurements

### Postural assessment scale for stroke patients (PASS)

The PASS assesses a participant's ability to maintain or change postures, from lying down to standing up. It consists of 12 items, seven mobility items and five balance items, each scored from 0 (unable to do) to 3 (normal performance). The maximum total score for the PASS is 36, with higher scores indicating better postural control. For each item, the researcher gave clear instructions, demonstrated the action if needed, and then observed the participant executing it. Participants were observed during each task, with safety aids provided as needed (*Chien et al., 2007*; *Estrada-Barranco et al., 2021*).

### Berg balance scale (BBS)

The BBS measures static and dynamic balance across 14 tasks, scored from 0 to 4, with a maximum score of 56. Tasks ranged from simple activities, like sitting unsupported, to more complex ones, like standing on one foot. The researcher provided clear verbal instructions for each task. Safety measures, including the use of a gait belt, were maintained throughout (*Badke et al., 2004*; *Barak & Duncan, 2006*).

### 6-minute walking distance (6MWD)

The 6MWD test required participants to walk at a comfortable pace for 6 min in a 30-m corridor. The total distance covered was measured to assess walking endurance. Safety was continuously monitored, and participants could use gait aids if needed. The test could be repeated if any inconsistencies or interruptions occurred during the first trial (*Kubo et al., 2018*).

### Functional ambulation categories scale (FAC)

The FAC rates walking ability on a scale from 0 (unable to walk) to 5 (fully independent). It assesses the level of assistance required during ambulation. A score of 0 indicates that the participant is unable to walk, while a score of 1 means that constant physical support is needed for balance, coordination, or weight-bearing. A score of 2 is given when occasional light touch is required for coordination or balance. Participants who can walk on even

terrain without hands-on aid but need supervision for safety or verbal guidance receive a score of 3. A score of 4 is assigned to those who can walk independently on flat surfaces but require oversight for more challenging situations, such as stairs or uneven terrain. Finally, a score of 5 indicates fully independent walking, where the participant can walk without assistance in all situations, including navigating stairs (*Mehrholz et al., 2007*).

## Statistical analysis

All analyses were conducted using MedCalc software (version 19.1.5, MedCalc Software Ltd, Ostend, Belgium), with an alpha level of 0.05 for statistical significance. Group differences in demographics were assessed using independent t-tests, chi-squared tests, and Mann-Whitney tests. The predictive accuracy of the PASS and BBS for community walking ability was evaluated through receiver operating characteristic (ROC) curves, with the area under the curve (AUC) used to differentiate home and community ambulators. AUC values were interpreted as follows: high accuracy (AUC > 0.90), moderate accuracy (AUC 0.70–0.90), low accuracy (AUC 0.50–0.69), and random guess (AUC < 0.50) (*Greiner, Pfeiffer & Smith, 2000*). Optimal cut-off scores for the BBS and PASS were identified using Youden's Index (sensitivity + specificity − 1) (*Ruopp et al., 2008*). To enhance the robustness of the cut-off values, the bias-corrected and accelerated (BCa) bootstrap method was applied with 1,000 iterations. Bootstrap confidence intervals were calculated for the AUC, Youden Index, and cut-off scores to improve the reliability of the findings. Diagnostic accuracy was further assessed using likelihood ratios (LR+ and LR-), where higher LR+ (>10) and lower LR- (<0.1) indicated greater clinical relevance. The interpretation of LR values was: Very useful (LR+ > 10, LR- < 0.1), moderate utility (LR+ 5–10, LR- 0.1–0.2), limited utility (LR+ 2–5, LR- 0.2–0.5), and minimal utility (LR+ 1–2, LR- 0.5–1) (*Deeks & Altman, 2004*; *Parikh et al., 2009*).

# RESULTS

Of the 47 participants enrolled, 46 completed both admission and discharge assessments. One participant dropped out due to a COVID-19 infection at the time of discharge, and no outcome was recorded. This dropout was unrelated to the study or rehabilitation program and is unlikely to have introduced bias into the results. Thus, 46 patients completed the study. Of these, 18 were classified as community ambulators and 28 as home ambulators. Baseline demographic and clinical characteristics of participants are presented in Table 1. The characteristics include variables such as age, gender, time since stroke onset, type of stroke, affected side, number of rehabilitation sessions, and Modified Rankin Scale (mRS) scores to reflect stroke severity. The majority of participants had moderate to severe disability at baseline, with 47.8% at mRS level 4 and 15.2% at mRS level 5, indicating significant functional limitations.

Mean scores for the PASS and BBS assessments showed significant differences between community and home ambulators, with community ambulators scoring higher on both scales. Details of the mean scores, standard deviations, and statistical comparisons are provided in Table 2.

**Table 1 Baseline characteristics of parcipants stratified by walking ability outcome (community *vs.* home ambulators).**

| Characteristics | Total (*n* = 46) |
|---|---|
| **Age;** (years); mean ± SD | 58.52 ± 11.16 |
| **Sex**; *n* (%) | |
| *Male* | 39 (84.8%) |
| *Female* | 7 (15.2%) |
| **Time since stroke onset;** (days); median (IQR) | 75.50 (68.00–148.00) |
| **Type of stroke;** *n* (%) | |
| *Ischemic stroke* | 37 (80.4%) |
| *Hemorrhagic stroke* | 9 (19.6%) |
| **Affected side;** *n* (%) | |
| *Left* | 20 (43.5%) |
| *Right* | 26 (56.5%) |
| **Number of visits;** median (IQR) | 16 (16-16) |
| **Modified Rankin Scale (mRS);** *n* (%) | |
| *2* | 6 (13.0%) |
| *3* | 11 (23.9%) |
| *4* | 22 (47.8%) |
| *5* | 7 (15.2%) |

**Table 2 Comparison of Postural Assessment Scale for Stroke Patients (PASS) and Berg Balance Scale (BBS) scores between community and home ambulators.**

| | Community ambulators (*n* = 18) | | Home ambulators (*n* = 28) | | | | |
|---|---|---|---|---|---|---|---|
| | Mean | SD | Mean | SD | Difference | 95% CI | *P* |
| **PASS** | 32.56 | 2.38 | 21.79 | 6.09 | −10.77 | [−13.81 to −7.73] | <0.001* |
| **BBS** | 50.78 | 2.16 | 23.43 | 14.75 | −27.35 | [−34.43 to −20.27] | <0.001* |

Notes:
PASS, Postural Assessment Scale for Stroke Patients; BBS, Berg Balance Scale.
* Statistical significance at *p* < 0.05.

Both the PASS and BBS demonstrated high accuracy in predicting community walking abilities, with AUC values of 0.955 (95% CI [0.850–0.994]) and 0.991 (95% CI [0.906–1.000]), respectively, indicating excellent discriminative power (Fig. 1 and Table 3). The optimal cut-off scores were ≥28 for PASS and ≥46 for BBS, both showing high sensitivity (94.44%, 95% CI [72.7–99.9]) and specificity values of 85.71% (95% CI [67.3–96.0]) for PASS and 96.43% (95% CI [81.7–99.9]) for BBS. The BBS had a stronger specificity, suggesting it may be slightly more reliable for identifying community ambulators. The positive likelihood ratios (LR+) of 6.61 (95% CI [2.65–16.49]) for PASS and 26.44 (95% CI [3.85–181.81]) for BBS indicate that BBS has a greater utility in confirming community ambulation. Both scales had negative likelihood ratios (LR-) below 0.1, emphasizing their effectiveness in ruling out non-community ambulators, thus

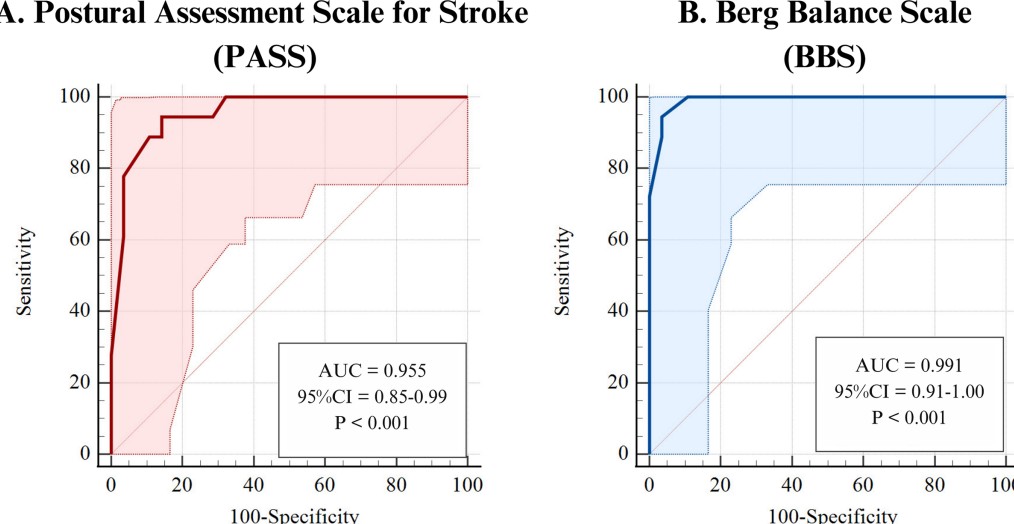

**Figure 1 Receiver operating characteristic (ROC) curves for predicting community walking ability using the (A) Postural Assessment Scale for Stroke Patients (PASS) and (B) the Berg Balance Scale (BBS).** The ROC curve for the BBS demonstrates high predictive accuracy with an Area Under the Curve (AUC) of 0.991, a cut-off score of ≥46, 94.4% sensitivity (correctly identifying community ambulators), and 96.4% specificity (correctly identifying home ambulators). The ROC curve for the PASS shows an AUC of 0.955, a cut-off score of ≥28, 94.4% sensitivity, and 85.7% specificity. Both tools demonstrate excellent accuracy in distinguishing community from home ambulators, with AUC values close to 1.0 indicating strong predictive performance and clinically relevant thresholds for walking ability classification.

supporting their strong clinical decision-making utility. Further details on these metrics can be found in Table 3.

Analysis of floor and ceiling effects showed that neither the PASS nor the BBS had significant floor effects, with both scoring 0%. This suggests that both scales are effective in evaluating lower levels of postural control and balance ability. The PASS had a slight ceiling effect of 4.35%, well below the 20% threshold that would indicate a significant issue. The BBS had no ceiling effect (0%), suggesting it may be more effective in distinguishing higher levels of balance performance.

A pairwise comparison of ROC curves showed that the AUCs of PASS and BBS differed by 0.04, with a standard error of 0.02. The 95% confidence interval for this difference ranged from −0.01 to 0.08, yielding a z statistic of 1.68, which was not statistically significant ($p = 0.093$). This result indicates that both instruments perform similarly in predicting community ambulation among patients with subacute stroke after rehabilitation. High AUC values and favorable likelihood ratios further support the utility of PASS and BBS for predicting walking ability at discharge.

## DISCUSSION

This study demonstrated that both the BBS and PASS are effective tools for predicting walking ability at discharge in patients with subacute stroke undergoing rehabilitation, consistent with prior research on their role in balance assessment and functional outcome

**Table 3 Area under the curve (AUC), cut-off score, sensitivity, specificity, positive likelihood ratio (LR+), and negative likelihood ratio (LR-).** AUC, cut-off score, sensitivity, specificity, LR+, and LR- of Postural Assessment Scale for Stroke Patients (PASS) and Berg Balance Scale (BBS) for predicting community walking in subacute stroke patients at discharge from the rehabilitation ward.

|  | PASS | BBS |
| --- | --- | --- |
| **AUC** | 0.955 | 0.991 |
|  | **High accuracy** | **High accuracy** |
| **Standard error** | 0.027 | 0.008 |
| 95% CI | [0.850–0.994] | [0.906–1.000] |
| **95% Bootstrap CI[a]** | [0.859–0.991] | [0.945–1.000] |
| **P-value** | <0.0001* | <0.0001* |
| **Youden index J** | 0.802 | 0.909 |
| 95% Bootstrap CI[a] | 0.575 to 0.909 | 0.750 to 0.964 |
| **Cut-off score** | > 28 | > 46 |
| 95% Bootstrap CI[a] | > 23 to > 30 | > 45 to > 49 |
| **True Positive (TP)** (n) | 17 | 17 |
| **True Negative (TN)** (n) | 24 | 27 |
| **False Positive (FP)** (n) | 4 | 1 |
| **False Negative (FN)** (n) | 1 | 1 |
| **Sensitivity** (95% CI) | 94.44 [72.7–99.9] | 94.44 [72.7–99.9] |
| **Specificity** (95% CI) | 85.71 [67.3–96.0] | 96.43 [81.7–99.9] |
| **LR+** (95% CI) | 6.61 [2.65–16.49] | 26.44 [3.85–181.81] |
|  | **Moderate utility** | **Very useful** |
| **LR-** (95% CI) | 0.07 [0.01–0.44] | 0.06 [0.01–0.39] |
|  | **Very useful** | **Very useful** |

Notes:
AUC, Area under the curve (AUC); LR+, positive likelihood ratio; LR-, negative likelihood ratio; PASS, Postural Assessment Scale for Stroke Patients; BBS, Berg Balance Scale (BBS).
* Statistical significance at *p* < 0.05.
[a] Bias-corrected and accelerated (BCa) bootstrap confidence interval (1,000 iterations; random number seed: 978).
Bold values indicate key diagnostic interpretations.

prediction. While both tools demonstrated high overall accuracy, the BBS showed slightly higher specificity. This specificity can be attributed to its inclusion of advanced balance tasks that assess critical balance and coordination abilities necessary for community ambulation.

The BBS, with its high specificity (93.33%) and a robust LR+ of 26.44, was particularly effective at identifying community ambulators. Patients scoring above the cut-off of 46 were 26 times more likely to achieve community ambulation, highlighting the BBS as a valuable clinical tool. This specificity can be attributed to the advanced balance tasks included in the BBS, which assess critical balance and coordination abilities necessary for community ambulation (*Kim & Oh, 2023*). For instance, tandem stance challenges the patient's ability to maintain balance with a narrow base of support, mimicking scenarios such as walking in crowded spaces. Similarly, single-leg stance requires significant lower limb strength and trunk control, which are crucial for tasks like stair climbing or recovering from balance disturbances (*Delfa-de-la-Morena et al., 2024*; *Labanca et al.,*

*2021*). These tasks collectively provide a comprehensive evaluation of both static and dynamic balance abilities, making the BBS highly effective for identifying community ambulators. However, these advanced tasks may also lead to floor effects, limiting the BBS's applicability for patients with severe impairments.

In contrast, the PASS focuses on fundamental postural control and includes tasks such as sitting without support, rolling, and transitioning from sitting to standing (*Estrada-Barranco et al., 2021*). These foundational tasks are particularly effective for evaluating patients with severe impairments, as they target core stability and postural control essential for recovery, including those in the early subacute phase (1 week to 3 months post-stroke) when neuroplasticity is heightened. For example, the "rolling" task assesses the ability to initiate movement and maintain control while changing positions, which is critical in the early recovery stages. "Sitting without support" measures core stability, enabling patients to safely perform seated tasks and progress to more advanced activities. Additionally, "transitioning from sitting to standing" evaluates lower limb strength and balance, a key milestone in regaining mobility. The inclusion of such basic tasks makes PASS sensitive to capturing small but clinically meaningful improvements in balance. Moreover, tasks like "standing without support" and "standing on one leg" allow the PASS to assess more advanced postural skills, bridging early and moderate recovery stages.

The findings from this study are consistent with previous research, including *Louie & Eng (2018)*, who also demonstrated the predictive value of the BBS for walking improvement in patients with stroke. However, our study offers several novel contributions compared to their work. Employing a prospective design, our research enabled real-time data collection and monitoring of patients with stroke during their rehabilitation journey, reducing biases inherent in retrospective analyses and strengthening predictive validity. Unlike Louie and Eng's study, which evaluated only the BBS, we directly compared the BBS and PASS, providing a head-to-head assessment of two commonly used tools to predict walking ability at discharge. This dual assessment offers clinicians a more comprehensive understanding of the relative effectiveness of each tool and their applicability for patients with varying functional levels. Furthermore, we used a combined outcome measure of the 6MWD and FAC at discharge to classify community walkers, integrating both endurance and functional mobility to provide a more nuanced understanding of walking readiness—factors not addressed in Louie and Eng's study.

Compared to *Wang, Chen & Wang (2022)*, our study provides a more immediate evaluation of walking ability. *Wang, Chen & Wang (2022)* assessed walking status 3 months post-stroke using telephone interviews, a method that, while valuable, is limited by its reliance on subjective reporting and delayed outcome collection. By contrast, our study assessed walking ability at discharge using objective measures, including the 6MWD and FAC. This approach offers a real-time assessment of walking capability, which is critical for tailoring post-discharge care plans and predicting readiness for community ambulation. Furthermore, *Wang, Chen & Wang (2022)* emphasized the correlation between PASS scores at admission and walking status at 3 months post-stroke, underscoring the importance of early balance assessments. Our findings expand on this by directly

comparing PASS with BBS and highlighting their complementary roles across different recovery stages.

We also addressed concerns regarding ceiling and floor effects for these scales. While earlier research reported such effects for both PASS and BBS (*Benaim et al., 1999*; *Blum & Korner-Bitensky, 2008*; *Mao et al., 2002*), we found no significant floor effects for either tool (0%). The PASS had a slight ceiling effect (4.35%), well below the 20% threshold of concern, whereas the BBS had no ceiling effect. These findings suggest that both scales effectively measure a wide range of abilities, with the BBS offering a slight advantage in distinguishing higher levels of balance performance. The absence of significant floor effects highlights the suitability of these tools for severely impaired patients, particularly the PASS, while the minimal ceiling effects ensure their utility in capturing progress across recovery stages.

In clinical practice, the choice between PASS and BBS should be guided by the patient's recovery stage and functional ability. PASS's simplicity and minimal equipment requirements make it ideal for resource-limited settings or for evaluating patients with severe impairments and early subacute phase of stroke survivors. The trunk control items of the PASS (Item 1: sitting without support, Item 6: supine to affected side lateral, Item 7: supine to non-affected side lateral, Item 8: supine to sitting up on the edge of the table, and Item 9: sitting on the edge of the table to supine) have been shown to predict activities of daily living (ADL) function 1 year after stroke (*Wang et al., 2005*). However, the PASS-Trunk Control (PASS-TC) exhibited a notable ceiling effect at 30% of the sample and had limited responsiveness beyond the first 30 days post-stroke, suggesting that it is most effective for early recovery assessment rather than long-term evaluation. In contrast, the BBS provides a more detailed assessment of balance abilities required for community ambulation and may be better suited for discharge planning or outpatient rehabilitation. Together, these tools offer a continuum of assessment: PASS is effective for capturing early recovery progress, while BBS is more appropriate for evaluating readiness for community ambulation. The assessment of trunk balance using the PASS can detect small but clinically meaningful changes in postural control, even in patients with severe impairments or highly deteriorated postural control. Their sequential use allows clinicians to tailor rehabilitation strategies to individual patient needs, optimizing outcomes.

Our study closed a gap in previous research by directly comparing PASS and BBS in the subacute phase of stroke recovery upon discharge from rehabilitation. This head-to-head comparison highlights the practical implications of our findings, emphasizing both tools' effectiveness. Although the BBS showed slightly better performance with a higher LR+ (26.44) than PASS (6.61), PASS's ease of use and shorter administration time (10–15 min) make it a practical choice in clinical settings where time efficiency is crucial. In contrast, the BBS requires 20–30 min to administer, which may be less convenient in busy environments.

## Strengths and limitations

The prospective cohort design of this study reduced biases and allowed for precise tracking of changes over time, offering a clearer observation of outcomes. Using the FAC and

6MWD tests at discharge provided objective assessments of community walking. Comparing the PASS and BBS offered insights into their practical applications and effectiveness in predicting community ambulation in patients with subacute stroke. However, this study was conducted at a single rehabilitation hospital, which may limit the generalizability of the findings to other rehabilitation settings. One limitation of this study is the exclusion of the Mini-BESTest, which is widely used to assess dynamic balance and fall risk. Unlike PASS, which is tailored for stroke rehabilitation, Mini-BESTest evaluates multiple balance components, including anticipatory and reactive postural control, which may provide additional insights into functional mobility. While PASS was chosen due to its specificity in postural control assessment for patients with stroke, future studies should incorporate Mini-BESTest to explore its predictive accuracy in post-stroke ambulatory outcomes. Moreover, results specific to subacute stroke may not apply to other recovery stages. Longer follow-up studies could better assess the sustainability of walking abilities over time.

## CONCLUSIONS

The high predictive accuracy of the PASS and BBS suggests that clinicians can effectively use these tools to evaluate and predict community walking ability after rehabilitation. Cut-off scores of >28 for PASS and >46 for BBS distinguish between those likely to achieve community ambulation and those suited for home ambulation. The PASS's ease of use makes it ideal for quick assessments, while the BBS's higher LR+ and accuracy support its use for more detailed evaluations. Clinicians should consider patient needs and clinical context when selecting the most appropriate tool to optimize rehabilitation outcomes.

## ACKNOWLEDGEMENTS

The authors acknowledge the use of artificial intelligence (AI), specifically OpenAI's ChatGPT, for improving the clarity and language of the manuscript. The AI tool was used solely for language refinement, and all scientific content, data analysis, and conclusions remain the responsibility of the authors.

### Funding

This work was supported by the 90th Anniversary of Chulalongkorn University Fund (Ratchadaphiseksomphot Endowment Fund). The funders had no role in study design, data collection and analysis, decision to publish, or preparation of the manuscript.

### Grant Disclosures

The following grant information was disclosed by the authors:
90th Anniversary of Chulalongkorn University Fund (Ratchadaphiseksomphot Endowment Fund).

### Competing Interests

The authors declare that they have no competing interests.

## Author Contributions

- Chutipa Worraridthanon conceived and designed the experiments, performed the experiments, analyzed the data, authored or reviewed drafts of the article, and approved the final draft.
- Maria Justine analyzed the data, authored or reviewed drafts of the article, and approved the final draft.
- Akkradate Siriphorn conceived and designed the experiments, analyzed the data, prepared figures and/or tables, authored or reviewed drafts of the article, and approved the final draft.

## Human Ethics

The following information was supplied relating to ethical approvals (*i.e.*, approving body and any reference numbers):

The Sirindhorn National Medical Rehabilitation Institute's Subcommittee on Human Research Ethics

## Data Availability

The raw measurements are available in the Supplemental File.

## Supplemental Information

Supplemental information for this article can be found online at http://dx.doi.org/10.7717/peerj.19322#supplemental-information.

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
