# Peer review of "Comparing the Postural Assessment Scale for Stroke and Berg Balance Scale for predicting community walking ability at discharge in subacute stroke: a prospective cohort study"

_PeerJ, doi:10.7717/peerj.19322_

## Round 0.1 · original submission · Major Revisions

Please respond to the comments from both reviewers. Note: The comments from R2 are mainly in their annotated PDF

·

Basic reporting

The core outcomes of the balance scale are BBS and Mini-Balance Evaluation Systems Test (Mini-BESTest). Why don't you emphasize in the introduction why PASS was selected for this study? It is recommended that Mini-BESTest be discussed at the limitations of the study.

It is preferable to describe the participant as "in patients with stroke" rather than "stroke patients"

Experimental design

Add information about the participant's ability to walk at the time of admission. If someone had "Community ambulators" from the time of admission, the results may not be valid.

Internal validation of the cut-off value by the bootstrap method is recommended. This enhances the validity of the results.

It would be nice to have True Positive (FP), True Negative (TN), Falles Positive (FP), and Falles Negative (FN) results when other researchers try to validate your cutoff. It is recommended that you add it.

Please add any information on the severity of the stroke in Table 1.

Validity of the findings

Sufficient information was described.

Reviewer 2 ·

Basic reporting

Some references should be updated

Experimental design

Original primary research within aims and scope of the journal

Validity of the findings

All underlying data have been provided, they are robust, statiscally sound and controlled.

Additional comments

The additional comment to author are as comments in the attached manuscript

Annotated reviews are not available for download in order to protect the identity of reviewers who chose to remain anonymous.

---

## Round 0.2 · accepted · Accept

Dear Authors
Thank you for making all the suggested changes
Congratulations!

·

Basic reporting

Thank you for revising the paper appropriately. No additional comments.

Experimental design

Thank you for revising the paper appropriately. No additional comments.

Validity of the findings

Thank you for revising the paper appropriately. No additional comments.

Reviewer 2 ·

Basic reporting

It is correct

Experimental design

It is correct

Validity of the findings

It is correct

Additional comments

The authors have made the changes suggested by the reviewers.